# Validation of the Portuguese Version of the Scleroderma Health Assessment Questionnaire

**DOI:** 10.3390/ijerph20227062

**Published:** 2023-11-14

**Authors:** Inês Genrinho, Pedro L. Ferreira, Tânia Santiago, Adriana Carones, Carolina Mazeda, Anabela Barcelos, Tiago Beirão, Flávio Costa, Inês Santos, Maura Couto, Maria Rato, Georgina Terroso, Paulo Monteiro

**Affiliations:** 1Rheumatology Department, Tondela Viseu Hospital Centre, 3504-509 Viseu, Portugal; 8257@hstviseu.min-saude.pt (I.S.); maura.couto.6098@hstviseu.min-saude.pt (M.C.); pjr_monteiro@hotmail.com (P.M.); 2Rheumatology Department, Baixo Vouga Hospital Centre, 3810-164 Aveiro, Portugal; 13194@chbv.min-saude.pt (I.G.); carolina.pereira.71522@chbv.min-saude.pt (C.M.); anabela.barcelos.11287@chbv.min-saude.pt (A.B.); 3Centre for Health Studies and Research, Faculty of Economics, University of Coimbra, 3004-512 Coimbra, Portugal; 4Rheumatology Department, Hospital and University Centre of Coimbra, 3004-512 Coimbra, Portugal; 12811@chuc.min-saude.pt (T.S.); adriana_carones92@hotmail.com (A.C.); 5EpiDoC Unit, CEDOC, NOVA Medical School, NOVA University of Lisbon, 1169-056 Lisbon, Portugal; 6Comprehensive Health Research Center (CHRC), NOVA University of Lisbon, 1169-056 Lisbon, Portugal; 7Rheumatology Department, Vila Nova de Gaia Espinho Hospital Center, 4434-502 Vila Nova de Gaia, Portugal; tiago.beirao@chvng.min-saude.pt (T.B.); flavio.costa@chvng.min-saude.pt (F.C.); 8Rheumatology Department, Hospital Centre of São João, 4200-319 Porto, Portugal; maria.rato@chsj.min-saude.pt (M.R.); georgina.terroso@chsj.min-saude.pt (G.T.)

**Keywords:** systemic sclerosis, SHAQ, outcome assessment, quality of life

## Abstract

The Health Assessment Questionnaire Disability Index (HAQ-DI) was completed with five visual analog scales to assess systemic sclerosis (SSc) called Scleroderma HAQ (SHAQ). We performed a validation of the European Portuguese version of SHAQ for patients with SSc. Patients with different forms of SSc from five Hospital Centers were invited. The reliability of the Portuguese SHAQ was evaluated by internal consistency and by test–retest reliability. Content validity was checked by two rheumatologists and by a panel of patients. Construct validity was assessed by structural validity and by known-groups hypothesis tests. Criterion validity was addressed with selected dimensions from the UCLA GIT 2.0, the SF-36v2, and the EuroQoL EQ-5D-5L. A total of 102 SSc patients agreed to participate, 31 of which answered to the retest. HAQ-DI demonstrated high internal consistency reliability (α = 0.866) and SHAQ also showed high test–retest reliability (ICC 0.61–0.95). We evidenced the unidimensionality of all VASs. HAQ-DI scores were worse in males, patients older than 65 years, and individuals with a diffuse form of SSc. Criterion validity was mainly evidenced through the correlation between the HAQ-DI and SF-36v2 physical summary measure (r = −0.688) and EQ-5D-5L index score (r = −0.723). Likewise, the SHAQ overall disease severity VAS was also correlated with SF-36v2 physical summary measure (r = −0.628). Mental score correlations were smaller. With the exception of the Raynaud’s VAS, all the other VASs correlated well with similar clinical variables. This paper provides evidence to demonstrate how reliable and valid the European Portuguese version of SHAQ is, to be used in SSc patients to assess the clinical severity under the perspective of patients.

## 1. Introduction

Systemic sclerosis (SSc), also known as scleroderma, is an immune-mediated rheumatic disease characterized by excessive collagen deposition in the skin and other systems, including the musculoskeletal, cardiopulmonary, renal, and gastrointestinal tract. Women are predominantly affected, with a female-to-male ratio of 3–8:1, specially in limited cutaneous SSc (lcSSc) form, while the diffuse type affects males and females at more comparable rates. The peak of incidence for the first symptoms usually arises between 45 and 64 years old [1,2].

Heterogeneity in the clinical presentation of SSc patients may induce different outcomes in the disease course. In some cases, disease may be stabilized across several months, while in other patients, particularly in the diffuse cutaneous SSC (dcSSc) subtype, the disease can have a fulminant clinical course. Treatment options are prescribed according to disease manifestations, with immunosuppressive (IS) drugs being the standard of care [3].

Due to multiorgan involvement, this pathology is associated with a significant impairment of health-related quality of life (HRQoL) and a high morbimortality [4,5]. The evaluation of the severity and activity caused by this disease usually requires several clinical examinations. Self-administered patient-reported outcomes (PROs) in health status questionnaires arise as a practical, inexpensive, reliable, and valid method to assess functional repercussions related to some rheumatic diseases and their impact on patients’ perspectives. 

Initially, the Health Assessment Questionnaire Disability Index (HAQ-DI) was developed for rheumatoid arthritis [6,7], but it has been proven to be a valuable tool to predict and assess outcomes in SSc disease. So, in 1991, Poole and Steen [8,9] added five specific visual analog scales (VASs) to address overall disease severity, Raynaud’s phenomenon, digital tip ulcers, and gastrointestinal and lung symptoms, creating a more disease-specific measure for SSc, the Scleroderma HAQ (SHAQ). The SHAQ is an accurate and feasible multisystem-specific tool to measure disease status changes that has been widely used in SSc [9]. This questionnaire was translated and validated in several languages, such as Brazilian Portuguese [10,11], Chinese [12], French [13], Italian [14], Japanese [15], Spanish [16], Swedish [17], and Turkish [18], but not yet to European Portuguese. Thus, the purpose of this study was to create and validate the SHAQ for Portuguese patients with SSc.

## 2. Materials and Methods

We followed the good practice principles to culturally adapt health outcome instruments to other linguistic contexts [19] and the Cosmin taxonomy [20]. This means that we tested the reliability (internal consistency and reproducibility) and the validity (content, construct, and criterion) of the obtained Portuguese version.

### 2.1. Cultural Adaptation and Content Validity

Before initiating this study, we contacted MAPI Research Institute to obtain permission to validate the Portuguese version of SHAQ. We received the information that a Portuguese non-validated version already existed, following the Food and Drug Administration (FDA) guidance on translation [21], and that we should use the version located on the MAPI website (https://eprovide.mapi-trust.org; accessed on 6 August 2021). 

This MAPI version was the result of the implementation of the forward–backward translation process; however, we still felt the necessity to perform a clinical review with two rheumatologists and a cognitive debriefing with patients to validate the content of this Portuguese version.

For the clinical review, a document was sent to the rheumatologists in which, for each item of the questionnaire, the English and Portuguese versions were placed, and the experts were asked to give us one of the following answers: (i) the translation is correct; (ii) the translation is wrong and an alternative is suggested; or (iii) if the translation is not incorrect, a proposed alternative would be better. Regarding the cognitive debriefing, two panels of five patients each were created. These panels approximately respected the age–sex joint distribution of patients with this pathology, prioritizing the lowest possible literacy. In each panel, patients were given the questionnaire to fill out, after which they were asked about the existence of missing, repeated, or ambiguous questions.

### 2.2. Participants

To validate the SHAQ, we then invited consecutive patients from five Portuguese Hospital Centers to participate in this study, between January and April 2022. Included patients fulfilled the 2013 American College of Rheumatology (ACR)/European League Against Rheumatism (EULAR) criteria for the classification of diffuse or limited SSc [22], combined with the EUSTAR (European Scleroderma Trial and Research Group) criteria for very early diagnosis of SSc (VEDOSS) [23], or presented SSc sine scleroderma. These patients were supposed to be autonomous, aged between 18 and 80 years, and have the ability to understand Portuguese and to grant informed consent. Pregnant women were excluded.

To test the reliability of the SHAQ, a smaller group of patients was also randomly selected to complete the measurement instrument a second time one month after the previous consultation.

The study was approved by the Ethics Committee of the Regional Health Authority of the Centre (ARSC 14/2020) and by the Ethics Committee from one of the main hospital centers (CHTV 05/16/09/2021). We also obtained authorization from all heads of the five rheumatology departments involved. Each participant signed a written consent form before filling out the questionnaire.

### 2.3. Measurement Instruments

Included patients were asked to complete sociodemographic, lifestyle, and clinical information; the Portuguese versions of the generic questionnaires to measure health status (SF-36v2) and quality of life (EQ-5D-5L); the University of California Los Angeles Scleroderma Clinical Trial Consortium Gastrointestinal Tract Instrument (UCLA GIT 2.0); and the SHAQ.

Regarding the sociodemographic variables, we collected data from sex, age, marital and employment status, and years of education. Also, smoking and alcoholism were proxies for lifestyle variables. Lastly, the clinical variables measured were the SSc subset classification, disease duration since diagnosis, 2013 ACR/EULAR classification criteria, and organ involvement. Immunosuppression was defined as exposure to at least one of the following: mycophenolate mofetil (MMF), cyclophosphamide (CYC), methotrexate (MTX), azathioprine (AZA), leflunomide, glucocorticoids (>10 mg/d prednisone-equivalent), rituximab, tocilizumab, and abatacept, for more than 6 months.The Short Form Health Survey (SF-36v2) is a generic instrument to measure the perception general population individuals have regarding their health status, on a scale from 0 (death) to 100 (perfect health status) [24]. It assesses eight dimensions (physical functioning—PF, bodily pain—BP, role limitations due to physical health—RP, general health perception—GH, mental health—MH, role limitations due to emotional problems—RE, vitality—VT, and social functioning—SF) and provides two component summary measures, a physical (PCS) and a mental one (MCS). In the case of the Portuguese version [25], these summary measures are normalized to the Portuguese general population.The EuroQoL EQ-5D-5L is a generic preference-based quality of life questionnaire that measures five dimensions (mobility, self-care, usual activities, pain/discomfort, and anxiety/depression) [26]. Each dimension has five levels of intensity. A visual analog scale EQ-VAS also asks for self-perception of general health status. Portuguese utilities can be computed by an algorithm based on general public preferences [27] and Portuguese norms are also available [28].The UCLA-GIT 2.0 questionnaire is a recognized and reliable instrument to assess gastrointestinal (GI) symptoms in SSc patients and its impact on quality of life [29]. Recently validated to the Portuguese population [30], it measures eight HRQoL dimensions (reflux, distention/bloating, fecal soilage, diarrhea, social functioning, emotional wellbeing, and constipation) and has been used in several clinical trials of GI treatments in patients with SSc as an outcome measure [31,32]. The total UCLA GIT score is calculated by averaging all the subscales, except the one for constipation, and ranges from 0 (best HRQoL) to 2.83 (worst HRQoL). The levels of GI severity symptoms used in this paper were described by the author [33].The SHAQ is comprised of the HAQ-DI plus six additional VASs—pain, intestinal, breathing, Raynaud’s, finger ulcer, and overall disease severity. The HAQ-DI contains 20 items and measures eight domains: (i) dressing and grooming, (ii) arising, (iii) eating, (iv) walking, (v) hygiene, (vi) reach, (vii) grip, and (viii) activities [34]. The answers for each question use a response scale from (0) without any difficulty to (3) unable to do. Intermediate response options are (1) with some difficulty and (2) with much difficulty. The highest score for any component question of each domain determines the score for that domain, with an exception for the necessity of aids or devices, where the score is automatically raised to two. A composite score is calculated by the average of the eight domains and ranges from 0 to 3, with a lower score indicating less impairment in function.Each additional VAS has a 1-week recall period, and it is represented by a line with a length of 10 cm. The value of the SHAQ VAS is multiplied by 0.3 to obtain the final score, ranging from 0 to 3 representing a minimum to maximum limitation, respectively.

### 2.4. Reliability

We tested the reliability of the Portuguese SHAQ version through internal consistency and intertemporal test–retest stability.

Internal consistency was tested through the score of Cronbach’s alpha coefficient, where accepted values should be between 0.70 and 0.90 [21]. The intertemporal stability was tested by the intraclass correlation coefficient (ICC) with two consecutive moments one month apart. We used the two-way mixed effects model and we looked at the absolute differences between the ratings of patients. We also followed the criteria that defend that an ICC lower than 0.50 corresponds to a weak correlation, between 0.50 and 0.75 and between 0.75 and 0.90 to a moderate and good one, respectively, and a score higher than 0.90 corresponds to an excellent correlation [35].

### 2.5. Validity

Construct validity tests included both structural validity and hypothesis testing with samples of sociodemographic and clinical variables. Criterion validity was then tested by comparing the HAQ-DI and SHAQ VAS scores with the scores obtained by the SF-36v2 and EQ-5D-5L [20].

To test structural validity, we conducted an exploratory factor analysis based on principal component estimates with a previous assessment of the sampling adequacy via the Kaiser–Meyer–Olkin (KMO) indicator and Bartlett’s test of sphericity. A KMO smaller than 0.50 or between 0.50 and 0.60 is considered unacceptable or poor, and scores between 0.60 and 0.70, between 0.70 and 0.80, between 0.80 and 0.90, or higher than 0.90 are seen as fair, average, good, or very good, respectively [36]. The significance of the Bartlett sphericity test should be smaller than 0.001 [37].

The hypothesis testing was performed with known sociodemographic (sex, age group) and clinical variable groups, based on the distribution of each HAQ-DI variable. Student’s *t*-test was used for two independent variables and ANOVA was used for more than two independent variables. 

To assess the criterion validity, we computed Pearson’s correlations between SHAQ items, SF-36v2 physical summary measures, and EQ-5D-5L index scores. We followed Cohen’s [38] rule from which correlations smaller than 0.30 are considered weak, between 0.30 and 0.50 are moderate, and higher than 0.50 are considered strong.

The statistical software used was SPSS v.28 (IBM, Armonk, NY, USA).

## 3. Results

### 3.1. Content Validity

Content validity was tested through a clinical review involving two rheumatologists and a cognitive debriefing conducted with two panels of five patients each. Both procedures did not result in any changes to the Portuguese version. That is, the Portuguese version of the SHAQ was accepted in clinical terms and accepted by patients without significant comment.

### 3.2. Sample

This study’s sample was composed of 102 SSc patients who immediately agreed to participate, a little bit higher than 100, the smallest size proposed by the Cosmin taxonomy [20]. Table 1 presents the sociodemographic and lifestyle behavior distributions, as well as the main clinical characteristics.

Our sample is mainly formed of females (82.4%) and patients older than 50 years (70.6%). The majority were married (69.6%) and had at least seven years of education (58.9%). In what concerns lifestyle behaviors, a small percentage were alcohol drinkers (15.7%) and an even smaller percentage were smokers (5.9%).

We also evidence that 62.7% were patients with limited SSc and 29.4% had a diffuse form. More than half of the patients (56.9%) had less than five years of disease duration, the Raynaud’s phenomenon was present in 94.1%, and abnormal nailfold capillary was present in 79.4% of the cases. In what concerns skin manifestations, sclerodactyly and telangiectasia were also prevalent at 63.7% and 62.7%, respectively. The autoantibodies profile showed a large majority of patients with positivity to ANA (89.2%), followed by anti-centromere (59.0%) and anti-topoisomerase I (19.6%).

Forty patients were under IS therapy, with MTX being the most frequent drug in 21.8%, followed by MMF in 11.9% and AZA in 5.9%. Glucocorticoid therapy was performed in only four patients.

Table 2 presents the generic health status and quality of life scores, as well as the UCLA GIT 2.0 domains and SHAQ scores.

It is evident that there are lower scores for the physical health status dimensions when compared to mental dimensions. In fact, the highest scores occur on the SF-36 dimensions of ’social role functioning’ (70.1) and ‘emotional role functioning’ (64.3). The overall quality of life was a little bit higher than average (66.8).

On the other hand, the total UCLA score presented mild severity with a mean of 0.39 ± 0.45, being the most frequently affected domains distension/bloating (mean 0.69 ± 0.78), constipation (mean 0.47 ± 0.64), and reflux (mean 0.46 ± 0.53). 

Concerning the SHAQ questionnaire, the worst visual analog scales in a descending order were overall disease severity (mean 35.2 ± 25.4), pain (mean 31.7 ± 22.8), and Raynaud’s (mean 26.2 ± 28.8), with a mean of 0.58 ± 0.51 for the HAQ-DI score.

### 3.3. Reliability

Thirty-one patients answered to the retest. The internal consistency of HAQ-DI was highly reliable (Cronbach’s α = 0.866) and ICC scores are presented in Table 3.

The lowest moderate ICC score was computed for the SHAQ Raynaud’s VAS. All the others can be classified as good. The ICC corresponding to the HAQ-DI can even be considered excellent.

### 3.4. Construct Validity

Before testing the structural validity through the factor analysis, we assessed the suitability of data for factor analysis. The KMO value was 0.798 higher than the recommended value of 0.6 and Bartlett’s test of sphericity was also associated with a significance <0.001. From factor analysis, and using the Kaiser criterion of eigenvalues higher than 1.0, we only obtained one factor corresponding to an eigenvalue of 3.073, evidencing the unidimensionality of all six SHAQ VASs. Looking at the corresponding component matrix, the VAS item with the highest loading was ‘overall disease severity’ (0.850), followed by ‘pain’ (0.766), ‘Raynaud’ (0.730), ‘gastrointestinal tract’ (0.719), ‘lung involvement’ (0.623), and ‘digital ulcers’ (0.571). Despite this unidimentionality, the five VASs will be analyzed separately.

To test construct validity, we compared the behavior of HAQ-DI with levels of selected sociodemographic and clinical variables (Table 4).

### 3.5. Criterion Validity

To test criterion validity, we correlated HAQ-DI and SHAQ-VAS scores with selected dimensions of UCLA GIT 2.0, SF-36v2, and EQ-5D-5L (Table 5).

This table evidences that HAQ-DI is mainly correlated with the SF-36v2’s physical summary measure, as well as with generic quality of life scores. In what concerns the more specific SHAQ VAS indicators, the physical summary measure is also highly correlated with the SHAQ overall disease severity VAS, and the SHAQ pain VAS was correlated with both the SF-36 Bodily Pain and EQ-5D-5L Pain/Discomfort dimensions. At last, the SHAQ intestinal VAS was highly correlated with UCLA total score and the Distension/bloating dimension.

Looking at clinical indicators, we investigated whether the scores obtained by the SHAQ VAS could be considered as determinants of relevant clinical variables (Table 6).

## 4. Discussion

Impairment in physical and psychosocial quality of life in patients with multisystem involvement associated with SSc is a major concern in clinical practice. Although, there is a lack of tools to evaluate disease activity, predict outcomes, and measure changes during the course of the disease. Self-administered questionnaires represent a standardized tool to evaluate consequences in daily life activities and assess patients’ perspectives of disease severity.

HAQ-DI internal consistency was shown to be highly reliable (α = 0.866). We also observed good test–retest reliability scores of the HAQ-DI and five of the SHAQ VASs (pain, intestinal, breathing, finger ulcers, and overall disease severity). The SHAQ Raynaud’s phenomenon VAS had an ICC less than the recommended 0.7, probably due to the large instability of this phenomenon, which can be dependent on external factors like climate and stress. A similar result has been found by other authors [12].

On the other hand, the most severe SSc subtype is represented by the diffuse cutaneous form with a large proportion of patients with multiorgan involvement and consequently a greater disability in daily living tasks and worse scores in the HAQ-DI [8]. Similar to other studies, and in what concerns the construct validity, we found higher-significance differences in HAQ-DI in patients with dcSSc [8,13,15]. In resemblance to some authors in previous validations, we did not find differences regarding disease duration since diagnosis [10,11,13,17]. This may be explained by most patients with a disease duration less than five years. Male sex and older patients (more than 65 years old) presented significant differences in HAQ-DI with worse scores. In previous studies, no differences were reported regarding gender. On the other hand, the Japanese also reported worse HAQ-DI scores in older patients [15].

Criterion validity was mainly evidenced through the correlation between the HAQ-DI and SF-36v2 physical summary measure (r = −0.688) and EQ-5D-5L index score (r = −0.723). Likewise, the SHAQ overall disease severity VAS was also correlated with the SF-36v2 physical summary measure (r = 0.628). Because SHAQ is a disability measure, mental score correlations were smaller [10,11,12,13,14,17,18]. With the exception of the Raynaud’s VAS, all the other VASs correlated well with similar clinical variables.

Although we have complied with the minimum sample size to validate a measurement instrument, we consider that it would be advantageous to replicate the study with a larger sample. Probably, this would provide us with better variability of the analyzed variables.

In particular, a larger sample size would increase the probability of recruiting participants with a longer duration of the disease. In this study, we collected consecutive patients from five Portuguese Hospital Centers to participate in this study, and almost 57% of these patients presented with less than 5 years of disease duration. Longer duration of disease might be associated with more complications and impaired quality of life; although, we believe that this does not affect the process of validation.

On the other hand, we know that the diffuse form of SSc is less prevalent in the population, occurring in about one-third to one-fourth of patients with systemic sclerosis. A larger sample might also include more patients with the diffuse subtype, but this difference in sample prevalence will still remain.

## 5. Conclusions

In conclusion, strictly following methodological criteria, the European Portuguese version of SHAQ has been shown to be reliable and valid to be used to describe the clinical severity of SSc from the perspective of patients.

## Figures and Tables

**Table 1 ijerph-20-07062-t001:** Sociodemographics, lifestyle behaviors, and clinical characteristics of patients (*n* = 102).

		*n*	%
Sex	Female	84	82.4
Age (years)	<50	30	29.4
	[50–65)	41	40.2
	≥65	31	30.4
	Minimum–Maximum	27–83	
	Median	56.5	
	Mean ± standard deviation	57.0 ± 12.5	
Marital status	Single	14	13.7
	Married/living with a partner	71	69.6
	Divorced/separated	10	9.8
	Widowed	7	6.9
Employment status	Employed	51	50.0
	Not employed	10	9.8
	Retired	37	36.3
	Housekeeper	4	3.9
Years of education	No formal education	2	2.0
	4 years	29	28.4
	5–6 years	11	10.8
	7–9 years	17	16.7
	10–12 years	32	31.4
	More than 12 years	11	10.8
Lifestyle behaviors	Smoker	6	5.9
	Alcohol drinker	16	15.7
Disease duration since diagnosis (years)	[0–5)	58	56.9
	≥5	44	43.1
	Minimum–Maximum	0–37	
	Median	4	
	Mean ± standard deviation	5.6 ± 6.0	
Subset classification of SSc	Diffuse	30	29.4
	Limited	64	62.7
	VEDOSS	7	6.9
	Sine scleroderma	1	1.0
2013 ACR/EULAR classification criteria	Skin thickening of the fingers of both hands	48	47.1
	Puffy fingers	28	27.5
	Sclerodactyly	65	63.7
	Digital tip ulcers	31	30.4
	Fingertip pitting scars	28	27.5
	Telangiectasia	64	62.7
	Abnormal nailfold capillary	81	79.4
	Pulmonary arterial hypertension	9	8.8
	Interstitial lung disease	22	21.6
	Raynaud’s phenomenon	96	94.1
Autoantibodies	ANA	91	89.2
	Anti-centromere	59	59.0
	Anti-topoisomerase I	20	19.6
	Anti-RNA polymerase III	1	1.0
	Other antibodies	12	12.7
Immunosuppression	Methotrexate (MTX)	22	21.8
	Mycophenolate mofetil (MMF)	12	11.9
	Azathioprine (AZA)	6	5.9
	Glucocorticoids (>10 mg/d prednisone-equivalent)	4	4

VEDOSS: Very Early Diagnosis of Systemic Sclerosis; ANA: Antinuclear Antibodies; SSc: systemic sclerosis; ACR: American College of Rheumatology; EULAR: European League Against Rheumatism.

**Table 2 ijerph-20-07062-t002:** Health status and quality of life variables.

		Min	Max	Mean	Sd
SF-36v2	Physical functioning (PF)	5.0	100.0	61.9	24.4
	Physical role functioning (RP)	0.0	100.0	57.0	31.4
	Bodily pain (BP)	0.0	100.0	49.7	22.3
	General health perceptions (GH)	0.0	87.0	41.4	19.4
	Vitality (VT)	10.0	100.0	43.7	24.1
	Social role functioning (SF)	0.0	100.0	70.1	28.6
	Emotional role functioning (RE)	0.0	100.0	64.3	29.4
	Mental health (MH)	4.0	100.0	60.1	27.7
	Physical component summary (PCS)	13.8	62.4	39.3	10.3
	Mental component summary (MCS)	22.0	70.2	47.5	12.1
EQ-5D-5L	EQ-5D Index	0.08	1.00	0.77	0.20
	EQ-VAS	4.0	100.0	66.8	18.7
UCLA	Reflux	0.00	2.50	0.46	0.53
	Distension/bloating	0.00	3.00	0.69	0.78
	Fecal soilage	0.00	3.00	0.16	0.48
	Diarrhea	0.00	2.00	0.36	0.54
	Social functioning	0.00	2.67	0.29	0.49
	Emotional wellbeing	0.00	2.89	0.37	0.63
	Constipation	0.00	3.50	0.47	0.64
	Total score	0.00	2.01	0.39	0.45
SHAQ	HAQ-DI	0.00	2.13	0.58	0.51
	Pain VAS	0.00	85.0	31.7	22.8
	Intestinal VAS	0.00	95.0	16.0	24.8
	Breathing VAS	0.00	84.0	17.4	24.6
	Raynaud’s VAS	0.00	100.0	26.2	28.8
	Finger ulcer VAS	0.00	100.0	18.3	27.8
	Overall disease severity VAS	0.00	90.0	35.2	25.4

Min: minimum; Max: maximum; Sd: standard deviation; VAS: visual analog scale; SF-36v2: Short Form Health Survey; EQ-5D-5L: The 5-level EQ-5D version; UCLA: University of California Los Angeles Scleroderma Clinical Trial Consortium Gastrointestinal Tract Instrument; SHAQ: Scleroderma Health Assessment Questionnaire Disability Index; HAQ-DI: Health Assessment Questionnaire Disability Index.

**Table 3 ijerph-20-07062-t003:** Reliability of SHAQ scales.

	ICC	ICC 95%CI
HAQ-DI	0.946	0.889–0.974
SHAQ Pain VAS	0.791	0.567–0.899
SHAQ Intestinal VAS	0.880	0.749–0.943
SHAQ Breathing VAS	0.743	0.466–0.876
SHAQ Raynaud’s VAS	0.608	0.187–0.811
SHAQ Finger ulcer VAS	0.818	0.624–0.912
SHAQ Overall disease severity VAS	0.868	0.726–0.936

SHAQ: Scleroderma Health Assessment Questionnaire Disability Index; HAQ-DI: Health Assessment Questionnaire Disability Index; VAS: visual analog scale; ICC: interclass correlation coefficient using the two-way mixed model with absolute agreement.

**Table 4 ijerph-20-07062-t004:** SHAQ total scores for different levels of sociodemographic and clinical variables.

		n	Mean	SD	|t| or F	*p*-Value
Sex	Female	84	0.53	0.48	2.204	0.030
	Male	18	0.82	0.57		
Age (years)	<50	30	0.50	0.09	3.245	0.043
	[50–65)	41	0.50	0.07		
	≥65	31	0.77	0.99		
Disease duration	[0–5]	58	0.52	0.49	1.447	0.151
	≥5	44	0.67	0.51		
Classification of SSc	Diffuse	30	0.81	0.60	2.653	0.011
	Not diffuse	72	0.50	0.43		

SSc: systemic sclerosis; SD: standard deviation; t: t-Student value; F: ANOVA value. HAQ-DI score presented significantly worse values in males (*p* = 0.030), patients older than 65 years old (*p* = 0.043), and in diffuse form (*p* = 0.011). No difference was found regarding disease duration (*p* = 0.151).

**Table 5 ijerph-20-07062-t005:** Correlations between SHAQ and other measures and clinical data.

		Correlation	*p*-Value
HAQ-DI	EQ-5D-5L	−0.723	<0.001
	EQ-VAS	−0.645	<0.001
	SF-36 Physical summary measure	−0.688	<0.001
	SF-36 Mental summary measure	−0.337	0.001
	UCLA total score	0.392	<0.001
SHAQ Pain VAS	SF-36 Bodily Pain	−0.629	<0.001
	EQ-5D-5L Pain/Discomfort	0.662	<0.001
SHAQ Intestinal VAS	UCLA Distension/bloating	0.690	<0.001
	UCLA Fecal soilage	0.454	<0.001
	UCLA Diarrhea	0.397	<0.001
	UCLA Constipation	0499	<0.001
	UCLA total score	0.762	<0.001
SHAQ Overall disease severity VAS	SF-36 Physical summary measure	−0.628	<0.001
	SF-36 Mental summary measure	−0.286	0.004
	EQ-5D-5L Index	−0.516	<0.001
	EQ-VAS	−0.549	<0.001

SHAQ: Scleroderma Health Assessment Questionnaire Disability Index; EQ-5D-5L: The 5-level EQ-5D version; HAQ-DI: Health Assessment Questionnaire Disability Index; UCLA: University of California Los Angeles Scleroderma Clinical Trial Consortium Gastrointestinal Tract Instrument; SF-36v2: Short Form Health Survey; VAS: visual analog scale.

**Table 6 ijerph-20-07062-t006:** SHAQ VAS and ACR/EULAR criteria.

SHAQ VAS	2013 ACR/EULAR Criteria		N	Mean	SD	|t|	*p*-Value
Breathing	Pulmonary arterial hypertension	No	93	14.8	2.2	3.547	<0.001
		Yes	9	43.7	10.4		
	Interstitial lung disease	No	80	14.7	22.3	2.107	0.038
		Yes	22	27.0	30.1		
Raynaud’s	Raynaud’s phenomenon	No	6	21.5	30.2	0.396	0.680
		Yes	96	26.5	58.7		
Finger ulcer	Fingertip ulcers	No	71	12.4	22.7	3.425	<0.001
		Yes	31	31.9	33.4		

SD: standard deviation; t: t-Student value; SHAQ: Scleroderma Health Assessment Questionnaire Disability Index; VAS: visual analog scale; ACR: American College of Rheumatology; EULAR: European League Against Rheumatism.

## Data Availability

The data underlying this article will be shared at reasonable request to the corresponding author.

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
