# Peer review of "Validation of the Portuguese Version of the Scleroderma Health Assessment Questionnaire"

_ijerph, 2023, doi:10.3390/ijerph20227062_

Round 1

Reviewer 1 Report

Comments and Suggestions for Authors

The manuscript by Genrinho et al. is a cross-sectional study aiming to validate the Portuguese version of SHAQ.

My only major objection concerns the duration of the disease in invited consecutive patients from five Portuguese Hospital Centers. Namely, as the authors explained in the discussion, in most patients the disease lasted for a short time, i.e. less than 5 years. It was necessary to increase the number of participance with a longer duration of the disease, since it is known that the duration of the disease, as well as the related complications and damage, which accumulate with the duration of the disease, can affect the quality of life. Also, a more even representation of individual scleroderma subtypes should have been achieved, as limited scleroderma dominates in the sample.

Author Response

Thank you for your comment. Following your remark, we included a sentence at the end of the discussion section:

“In particular, a larger sample size would increase the probability of recruiting participants with a longer duration of the disease. In this study, we collected consecutive patients from five Portuguese Hospital Centers to participate in this study, almost 57% of these patients presented less than 5 years of disease duration. Longer duration of disease might be associated with more complications and impaired quality of life, although, we believed that this doesn’t affect the process of validation.

On the other hand, we know that the diffuse form of SSc is less prevalent in the population, occurring in about one third to one fourth of patients with systemic sclerosis. A larger sample might also include more patients with the diffuse subtype, but this difference in sample prevalence will still remain.”

Reviewer 2 Report

Comments and Suggestions for Authors

I have no comments on the validation study. The authors followed the prescribed principles to culturally adapt health outcomes instruments to other linguistic contexts and the Cosmin taxonomy
The validation results are described correctly and clearly for the reader.
I would recommend adding a legend to the tables explaining any abbreviations that are used.

Comments on the Quality of English Language

I have no comments.

Author Response

Thank you for your remark. We have improved the legends to the tables 2, 3, 4, 5, and 6.

Reviewer 3 Report

Comments and Suggestions for Authors

This paper is a well-written and informative contribution to the study of psychometric properties of the Portuguese version of the Scleroderma Health Assessment Questionnaire (SHAQ).

The study design and data analysis appear to be rigorous and appropriate. This paper makes an important contribution to the literature that SHAQ is a useful measurement tool in other cultures.  

In general, the introduction is too brief and needs to include more important details. An overview of how SHAQ has yielded psychometric outcomes in other cultures, for instance, is lacking. It is crucial that the writers include an overview of the SHAQ literature in the introduction. What did the original study yield on the psychometrics of the measure? Did consequent studies replicate these results? Or was there any need for modification/item removal? All these are needed to be included in the Introduction. Have previous studies shown cross-cultural differences in SHAQ construct? Please also provide some hypotheses on what you expect about the factor structure and Portugese version of SHAQ and statistics that you have used to examine the validity of the SHAQ.

The result on the factor structure of SHAQ is worth elaborating.  The following sentence is too general: The unidimentionality if all six SHAQ VAS was guaranteed. Why only factor analysis on SHAQ VAS? Why was HAQ-DI left out? The correlations between the SHAQ-V scales and the HAQ-DI may also be of interest.

I see that there are more references to international results in the discussion, but it would be useful to put some of this in the introductory section, which could provide a basis for the research question. 

In my opinion, the above-mentioned additional notes and analysis would greatly increase the value of this study. Congratulations to the authors for an interesting and valuable study.

Author Response

Thank you for your comments.

We have rephrased the first part of the introduction:

“Systemic sclerosis (SSc), also known as scleroderma, is an immune-mediated rheumatic disease characterized by excessive collagen deposition in the skin and other systems, including the musculoskeletal, cardiopulmonary, renal, and gastrointestinal tract. Women are predominantly affected, with a female-to-male ratio of 3–8:1, specially in limited cutaneous SSc (lcSSc) form, while diffuse type affect males and females at more comparable rates.  The peak of incidence for the first symptoms usually arise between 45 and 64 years old [3,4].

Heterogeneity in clinical presentation of SSc patients may induce different outcomes in disease course. In some cases, disease may be stabilized across several months while in other patients, particularly in diffuse cutaneous SSC (dcSSc) subtype, disease can have a fulminant clinical course. Treatment options are prescribed according to disease manifestations, being the immunosuppressive (IS) drugs the standard of care. [xx].

Xx - Park R, Nevskaya T, Baron M, Pope J (2022) Canadian Scleroderma Research Group. Immunosuppression use in early systemic sclerosis may be increasing over time. J Scleroderma Relat Disord. 7(1): 33–41. doi: 10.1177/23971983211000971

Due to a multiorgan involvement, this pathology is associated to a significant impairment of health-related quality of life (HRQoL) and a high morbimortality [1,2]. The evaluation of the severity and activity caused by this disease usually requires several clinical examinations. Self-administered patients reported outcomes (PRO) in health status questionnaires arise as a practical, inexpensive, reliable, and valid method to assess functional repercussions related to some rheumatic diseases and their impact on patient’s perspectives.

Initially, the Health Assessment Questionnaire Disability Index (HAQ-DI) was developed for rheumatoid arthritis [5,6], but it has been proven to be a valuable tool to predict and assess outcomes in SSc disease. So, in 1991, Poole and Steen [7,8] added five specific Visual Analogue Scales (VAS) to address, respectively, overall disease severity, Raynaud’s phenomenon, digital tip ulcers, gastrointestinal and lung symptoms, creating a more disease-specific measure for SSc, the Scleroderma HAQ (SHAQ). The SHAQ is an accurate and feasible multisystem-specific tool to measure the disease status changes that has been widely used in SSc. [8].

This questionnaire was translated and validated into several languages, such as Brazilian Portuguese [9,10], Chinese [11], French [12], Italian [13], Japanese [14], Spanish [15], Swedish [16], and Turkish [17], but not yet to European Portuguese. Thus, the purpose of this study was to create and validate the SHAQ for Portuguese patients with SSc.

As mentioned in the paper, previously to the factor analysis on VAS SHAQ items, we computed KMO and Bartlett’s test of sphericity. Both criteria were met and only one factor was obtained.

To explain a little bit more, we included the following sentence under the section 3.3 Construct validity:

“From factor analysis, and using the Kaiser criterion of eigenvalues higher that 1.0, we only obtained one factor corresponding to an eigenvalue of 3.073, evidencing the unidimensionality of all six SHAQ VAS. Looking at the corresponding component matrix, the VAS item with higher loading was ‘overall disease severity’ (0.850), followed by ‘pain’ (0.766), ‘Raynaud’ (0.730), 'gastrointestinal tract’ (0.719), ‘lung involvement’ (0.623) and ‘digital ulcers’ (0.571).”

Reviewer 4 Report

Comments and Suggestions for Authors

I think your papers are very important for the clinical practice of SSc. I have a few questions.

1) In recent years, useful treatments for SSc, such as rituximab, have emerged. https://doi.org/10.1016/S2665-9913(21)00107-7

Please add treatment history to patient background.

2) The mRSS can be used to predict the severity of SSc as well as its cutaneous stiffness.Matsuda KM, Yoshizaki A, Kuzumi A, Fukasawa T, Ebata S, Miura S, Toyama T, Yoshizaki A, Sumida H, Asano Y, Oba K, Sato S. Skin thickness score as a surrogate marker of organ involvements in systemic sclerosis: a retrospective observational study. Arthritis Res Ther. 2019 May 28;21(1):129. doi: 10.1186/s13075-019-1919-6. PMID: 31138286; PMCID: PMC6540426.

mRSS should also be added to the patient background.

3) The recent advent of new autoantibody assays has made it possible to measure a large number of autoantibodies. The results of a comprehensive antibody assay using a new autoantibody assay may lead to a reconsideration of the diagnosis in some cases.

Norimatsu Y, Matsuda KM, Yamaguchi K, Ono C, Okumura T, Kogo E, Kotani H, Hisamoto T, Kuzumi A, Fukasawa T, Yoshizaki-Ogawa A, Goshima N, Sato S, Yoshizaki A The Autoantibody Array Assay: A Novel Autoantibody Detection Method. Diagnostics (Basel). 2023 Sep 13;13(18):2929. doi: 10.3390/diagnostics13182929. PMID: 37761295; PMCID: PMC10528021.

Provide details regarding other autoantibodies in the background. Also, please specify if there was no overlap syndrome.

Author Response

Thank you for your suggestion. We added the data on the immunosuppressive therapy that patients in the sample were taking (table 1). However, none of the patients were taking Rituximab, even though this drug was evaluated.

We also added a note in materials and methods section.

Thank you for your suggestion. The mRSS score is indeed quite relevant since the literature reports that higher scores are associated with worse HRQoL. However, since it is not one of the criteria for diagnosis, it was not performed in all patients and was not considered in this analysis. We will consider your suggestion in future projects.

Thank you for your suggestion. Although the new antibody patterns are reported in the literature, our sample showed a very small number of patients (12: 7 antiPm, 2 antiU3, 2 antiKu, and 1 antiU1 ) with Ac different from those considered in the ACR/EULAR 2013 criteria. For this reason, in table 1 they are represented as "other antibodies", and no robust conclusions can be drawn given the low prevalence. Again, since it is considered a PRO validation, we consider this topic to be outside the scope of the research question.

Again, in our sample we obtained very few patients with overlap syndrome and no ability to do detailed analyses.